# Practical Deep Learning with Bayesian Principles

**Kazuki Osawa,**[1] **Siddharth Swaroop,**[2,*] **Anirudh Jain,**[3,*,†] **Runa Eschenhagen,**[4,†]
**Richard E. Turner,**[2] **Rio Yokota,**[1] **Mohammad Emtiyaz Khan**[5,‡].

[1] Tokyo Institute of Technology, Tokyo, Japan
[2] University of Cambridge, Cambridge, UK
[3] Indian Institute of Technology (ISM), Dhanbad, India
[4] University of Osnabrück, Osnabrück, Germany
[5] RIKEN Center for AI Project, Tokyo, Japan

## Abstract

Bayesian methods promise to fix many shortcomings of deep learning, but they are impractical and rarely match the performance of standard methods, let alone improve them. In this paper, we demonstrate practical training of deep networks with natural-gradient variational inference. By applying techniques such as batch normalisation, data augmentation, and distributed training, we achieve similar performance in about the same number of epochs as the Adam optimiser, even on large datasets such as ImageNet. Importantly, the benefits of Bayesian principles are preserved: predictive probabilities are well-calibrated, uncertainties on out-of-distribution data are improved, and continual-learning performance is boosted. This work enables practical deep learning while preserving benefits of Bayesian principles. A PyTorch implementation[1] is available as a plug-and-play optimiser.

## 1 Introduction

Deep learning has been extremely successful in many fields such as computer vision [29], speech processing [17], and natural-language processing [39], but it is also plagued with several issues that make its application difficult in many other fields. For example, it requires a large amount of high-quality data and it can overfit when dataset size is small. Similarly, sequential learning can cause forgetting of past knowledge [27], and lack of reliable confidence estimates and other robustness issues can make it vulnerable to adversarial attacks [6]. Ultimately, due to such issues, application of deep learning remains challenging, especially for applications where human lives are at risk.

Bayesian principles have the potential to address such issues. For example, we can represent uncertainty using the posterior distribution, enable sequential learning using Bayes' rule, and reduce overfitting with Bayesian model averaging [19]. The use of such Bayesian principles for neural networks has been advocated from very early on. Bayesian inference on neural networks were all proposed in the 90s, e.g., by using MCMC methods [41], Laplace's method [35], and variational inference (VI) [18, 2, 49, 1]. Benefits of Bayesian principles are even discussed in machine-learning textbooks [36, 3]. Despite this, they are rarely employed in practice. This is mainly due to computational concerns, unfortunately overshadowing their theoretical advantages.

The difficulty lies in the computation of the posterior distribution, which is especially challenging for deep learning. Even approximation methods, such as VI and MCMC, have historically been difficult

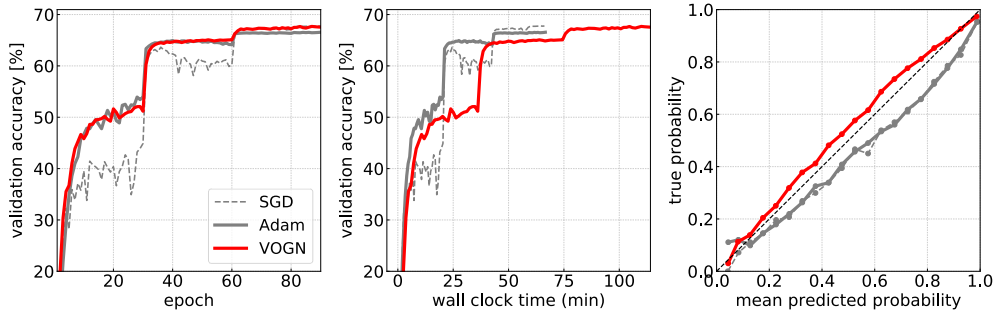

Figure 1: Comparing VOGN [24], a natural-gradient VI method, to Adam and SGD, training ResNet-18 on ImageNet. The two left plots show that VOGN and Adam have similar convergence behaviour and achieve similar performance in about the same number of epochs. VOGN achieves 67.38% on validation compared to 66.39% by Adam and 67.79% by SGD. Run-time of VOGN is 76 seconds per epoch compared to 44 seconds for Adam and SGD. The rightmost figure shows the calibration curve. VOGN gives calibrated predictive probabilities (the diagonal represents perfect calibration).

to scale to large datasets such as ImageNet [47]. Due to this, it is common to use less principled approximations, such as MC-dropout [9], even though they are not ideal when it comes to fixing the issues of deep learning. For example, MC-dropout is unsuitable for continual learning [27] since its posterior approximation does not have mass over the whole weight space. It is also found to perform poorly for sequential decision making [45]. The form of the approximation used by such methods is usually rigid and cannot be easily improved, e.g., to other forms such as a mixture of Gaussians. The goal of this paper is to make more principled Bayesian methods, such as VI, practical for deep learning, thereby helping researchers tackle its key limitations.

We demonstrate practical training of deep networks by using recently proposed natural-gradient VI methods. These methods resemble the Adam optimiser, enabling us to leverage existing techniques for initialisation, momentum, batch normalisation, data augmentation, and distributed training. As a result, we obtain similar performance in about the same number of epochs as Adam when training many popular deep networks (e.g., LeNet, AlexNet, ResNet) on datasets such as CIFAR-10 and ImageNet (see Fig. 1). The results show that, despite using an approximate posterior, the training methods preserve the benefits coming from Bayesian principles. Compared to standard deep-learning methods, the predictive probabilities are well-calibrated, uncertainties on out-of-distribution inputs are improved, and performance for continual-learning tasks is boosted. Our work shows that practical deep learning is possible with Bayesian methods and aims to support further research in this area.

**Related work.** Previous VI methods, notably by Graves [14] and Blundell et al. [4], require significant implementation and tuning effort to perform well, e.g., on convolution neural networks (CNN). Slow convergence is found to be especially problematic for sequential problems [45]. There appears to be no reported results with complex networks on large problems, such as ImageNet. Our work solves these issues by applying deep-learning techniques to natural-gradient VI [24, 56].

In their paper, Zhang et al. [56] also employed data augmentation and batch normalisation for a natural-gradient method called Noisy K-FAC (see Appendix A) and showed results on VGG on CIFAR-10. However, a mean-field method called Noisy Adam was found to be unstable with batch normalisation. In contrast, we show that a similar method, called Variational Online Gauss-Newton (VOGN), proposed by Khan et al. [24], works well with such techniques. We show results for distributed training with Noisy K-FAC on Imagenet, but do not provide extensive comparisons since tuning it is time-consuming. Many of our techniques can speed-up Noisy K-FAC, which is promising.

Many other approaches have recently been proposed to compute posterior approximations by training deterministic networks [46, 37, 38]. Similarly to MC-dropout, their posterior approximations are not flexible, making it difficult to improve the accuracy of their approximations. On the other hand, VI offers a much more flexible alternative to apply Bayesian principles to deep learning.

## 2 Deep Learning with Bayesian Principles and Its Challenges

The success of deep learning is partly due to the availability of scalable and practical methods for training deep neural networks (DNNs). Network training is formulated as an optimisation problem where a loss between the data and the DNN's predictions is minimised. For example, in a supervised learning task with a dataset $\mathcal{D}$ of $N$ inputs $\mathbf{x}_i$ and corresponding outputs $\mathbf{y}_i$ of length $K$, we minimise a loss of the following form: $\bar{\ell}(\mathbf{w}) + \delta \mathbf{w}^\top \mathbf{w}$, where $\bar{\ell}(\mathbf{w}) := \frac{1}{N} \sum_i \ell(\mathbf{y}_i, \mathbf{f}_w(\mathbf{x}_i))$, $\mathbf{f}_w(\mathbf{x}) \in \mathbb{R}^K$ denotes the DNN outputs with weights $\mathbf{w}$, $\ell(\mathbf{y}, \mathbf{f})$ denotes a differentiable loss function between an output $\mathbf{y}$ and the function $\mathbf{f}$, and $\delta > 0$ is the $L_2$ regulariser.[2] Deep learning relies on stochastic-gradient (SG) methods to minimise such loss functions. The most commonly used optimisers, such as stochastic-gradient descent (SGD), RMSprop [53], and Adam [25], take the following form[3] (all operations below are element-wise):

$$\mathbf{w}_{t+1} \leftarrow \mathbf{w}_t - \alpha_t \frac{\hat{\mathbf{g}}(\mathbf{w}_t) + \delta \mathbf{w}_t}{\sqrt{\mathbf{s}_{t+1}} + \epsilon}, \qquad \mathbf{s}_{t+1} \leftarrow (1 - \beta_t)\mathbf{s}_t + \beta_t \left(\hat{\mathbf{g}}(\mathbf{w}_t) + \delta \mathbf{w}_t\right)^2, \qquad (1)$$

where $t$ is the iteration, $\alpha_t > 0$ and $0 < \beta_t < 1$ are learning rates, $\epsilon > 0$ is a small scalar, and $\hat{\mathbf{g}}(\mathbf{w})$ is the stochastic gradients at $\mathbf{w}$ defined as follows: $\hat{\mathbf{g}}(\mathbf{w}) := \frac{1}{M} \sum_{i \in \mathcal{M}_t} \nabla_w \ell(\mathbf{y}_i, \mathbf{f}_w(\mathbf{x}_i))$ using a minibatch $\mathcal{M}_t$ of $M$ data examples. This simple update scales extremely well and can be applied to very large problems. With techniques such as initialisation protocols, momentum, weight-decay, batch normalisation, and data augmentation, it also achieves good performance for many problems.

In contrast, the full Bayesian approach to deep learning is computationally very expensive. The posterior distribution can be obtained using Bayes' rule: $p(\mathbf{w}|\mathcal{D}) = \exp\left(-N\bar{\ell}(\mathbf{w})/\tau\right) p(\mathbf{w})/p(\mathcal{D})$ where $0 < \tau \leq 1$.[4] This is costly due to the computation of the marginal likelihood $p(\mathcal{D})$, a high-dimensional integral that is difficult to compute for large networks. Variational inference (VI) is a principled approach to more scalably estimate an approximation to $p(\mathbf{w}|\mathcal{D})$. The main idea is to employ a parametric approximation, e.g., a Gaussian $q(\mathbf{w}) := \mathcal{N}(\mathbf{w}|\boldsymbol{\mu}, \boldsymbol{\Sigma})$ with mean $\boldsymbol{\mu}$ and covariance $\boldsymbol{\Sigma}$. The parameters $\boldsymbol{\mu}$ and $\boldsymbol{\Sigma}$ can then be estimated by maximising the *evidence lower bound (ELBO)*:

$$\text{ELBO:} \quad \mathcal{L}(\boldsymbol{\mu}, \boldsymbol{\Sigma}) := -N\mathbb{E}_q\left[\bar{\ell}(\mathbf{w})\right] - \tau\mathbb{D}_{KL}[q(\mathbf{w}) \,\|\, p(\mathbf{w})], \qquad (2)$$

where $\mathbb{D}_{KL}[\cdot]$ denotes the Kullback-Leibler divergence. By using more complex approximations, we can further reduce the approximation error, but at a computational cost. By formulating Bayesian inference as an optimisation problem, VI enables a practical application of Bayesian principles.

Despite this, VI has remained impractical for training large deep networks on large datasets. Existing methods, such as Graves [14] and Blundell et al. [4], directly apply popular SG methods to optimise the variational parameters in the ELBO, yet they fail to get a reasonable performance on large problems, usually converging very slowly. The failure of such direct applications of deep-learning methods to VI is not surprising. The techniques used in one field may not directly lead to improvements in the other, but it will be useful if they do, e.g., if we can optimise the ELBO in a way that allows us to exploit the tricks and techniques of deep learning and boost the performance of VI. The goal of this work is to do just that. We now describe our methods in detail.

## 3 Practical Deep Learning with Natural-Gradient Variational Inference

In this paper, we propose natural-gradient VI methods for practical deep learning with Bayesian principles. The natural-gradient update takes a simple form when estimating exponential-family approximations [23, 22]. When $p(\mathbf{w}) := \mathcal{N}(\mathbf{w}|0, \mathbf{I}/\delta)$, the update of the natural-parameter $\boldsymbol{\lambda}$ is performed by using the stochastic gradient of the *expected regularised-loss*:

$$\boldsymbol{\lambda}_{t+1} = (1 - \tau\rho)\boldsymbol{\lambda}_t - \rho\nabla_\mu\mathbb{E}_q\left[\bar{\ell}(\mathbf{w}) + \tfrac{1}{2}\tau\delta\mathbf{w}^\top\mathbf{w}\right], \qquad (3)$$

where $\rho > 0$ is the learning rate, and we note that the stochastic gradients are computed with respect to $\boldsymbol{\mu}$, the *expectation parameters* of $q$. The *moving average* above helps to deal with the stochasticity of the gradient estimates, and is very similar to the moving average used in deep learning (see (1)). When $\tau$ is set to 0, the update essentially minimises the regularised loss (see Section 5 in Khan et al. [24]). These properties of natural-gradient VI makes it an ideal candidate for deep learning.

Recent work by Khan et al. [24] and Zhang et al. [56] further show that, when $q$ is Gaussian, the update (3) assumes a form that is strikingly similar to the update (1). For example, the Variational Online Gauss-Newton (VOGN) method of Khan et al. [24] estimates a Gaussian with mean $\boldsymbol{\mu}_t$ and a diagonal covariance matrix $\boldsymbol{\Sigma}_t$ using the following update:

$$\boldsymbol{\mu}_{t+1} \leftarrow \boldsymbol{\mu}_t - \alpha_t \frac{\hat{\mathbf{g}}(\mathbf{w}_t) + \tilde{\delta}\boldsymbol{\mu}_t}{\mathbf{s}_{t+1} + \tilde{\delta}}, \quad \mathbf{s}_{t+1} \leftarrow (1 - \tau\beta_t)\mathbf{s}_t + \beta_t \frac{1}{M} \sum_{i \in \mathcal{M}_t} (\mathbf{g}_i(\mathbf{w}_t))^2, \qquad (4)$$

where $\mathbf{g}_i(\mathbf{w}_t) := \nabla_w \ell(y_i, f_{w_t}(\mathbf{x}_i))$, $\mathbf{w}_t \sim \mathcal{N}(\mathbf{w}|\boldsymbol{\mu}_t, \boldsymbol{\Sigma}_t)$ with $\boldsymbol{\Sigma}_t := \text{diag}(1/(N(\mathbf{s}_t + \tilde{\delta})))$, $\tilde{\delta} := \tau\delta/N$, and $\alpha_t, \beta_t > 0$ are learning rates. Operations are performed element-wise. Similarly to (1), the vector $\mathbf{s}_t$ adapts the learning rate and is updated using a moving average.

A major difference in VOGN is that the update of $\mathbf{s}_t$ is now based on a Gauss-Newton approximation [14] which uses $\frac{1}{M}\sum_{i \in \mathcal{M}_t}(\mathbf{g}_i(\mathbf{w}_t))^2$. This is fundamentally different from the SG update in (1) which instead uses the gradient-magnitude $(\frac{1}{M}\sum_{i \in \mathcal{M}_t}\mathbf{g}_i(\mathbf{w}_t) + \delta\mathbf{w}_t)^2$ [5]. The first approach uses the sum *outside* the square while the second approach uses it *inside*. VOGN is therefore a second-order method and, similarly to Newton's method, does not need a square-root over $\mathbf{s}_t$. Implementation of this step requires an additional calculation (see Appendix B) which makes VOGN a bit slower than Adam, but VOGN is expected to give better variance estimates (see Theorem 1 in Khan et al. [24]).

The main contribution of this paper is to demonstrate practical training of deep networks using VOGN. Since VOGN takes a similar form to SG methods, we can easily borrow existing deep-learning techniques to improve performance. We will now describe these techniques in detail. Pseudo-code for VOGN is shown in Algorithm 1.

**Batch normalisation:** Batch normalisation [20] has been found to significantly speed up and stabilise training of neural networks, and is widely used in deep learning. BatchNorm layers are inserted between neural network layers. They help stabilise each layer's input distribution by normalising the running average of the inputs' mean and variance. In our VOGN implementation, we simply use the existing implementation with default hyperparameter settings. We do not apply L2 regularisation and weight decay to BatchNorm parameters, like in Goyal et al. [13], or maintain uncertainty over the BatchNorm parameters. This straightforward application of batch normalisation works for VOGN.

**Data Augmentation:** When training on image datasets, data augmentation (DA) techniques can improve performance drastically [13]. We consider two common real-time data augmentation techniques: random cropping and horizontal flipping. After randomly selecting a minibatch at each iteration, we use a randomly selected cropped version of all images. Each image in the minibatch has a 50% chance of being horizontally flipped.

We find that directly applying DA gives slightly worse performance than expected, and also affects the calibration of the resulting uncertainty. However, DA increases the effective sample size. We therefore modify it to be $\rho N$ where $\rho \geq 1$, improving performance (see step 2 in Algorithm 1). The reason for this performance boost might be due to the complex relationship between the regularisation $\delta$ and $N$. For the regularised loss $\bar{\ell}(\mathbf{w}) + \delta\mathbf{w}^\top\mathbf{w}$, the two are unidentifiable, i.e., we can multiply $\delta$ by a constant and reduce $N$ by the same constant without changing the minimum. However, in a Bayesian setting (like in (2)), the two quantities are separate, and therefore changing the data might also change the optimal prior variance hyperparameter in a complicated way. This needs further theoretical investigations, but our simple fix of scaling $N$ seems to work well in the experiments.

We set $\rho$ by considering the specific DA techniques used. When training on CIFAR-10, the random cropping DA step involves first padding the 32x32 images to become of size 40x40, and then taking randomly selected 28x28 cropped images. We consider this as effectively increasing the dataset size by a factor of 5 (4 images for each corner, and one central image). The horizontal flipping DA step doubles the dataset size (one dataset of unflipped images, one for flipped images). Combined, this gives $\rho = 10$. Similar arguments for ImageNet DA techniques give $\rho = 5$. Even though $\rho$ is another hyperparameter to set, we find that its precise value does not matter much. Typically, after setting an estimate for $\rho$, tuning $\delta$ a little seems to work well (see Appendix E).

**Algorithm 1:** Variational Online Gauss Newton (VOGN)

---

1: Initialise $\boldsymbol{\mu}_0, \mathbf{s}_0, \mathbf{m}_0$.
2: $N \leftarrow \rho N, \tilde{\delta} \leftarrow \tau\delta/N$.
3: **repeat**
4:     Sample a minibatch $\mathcal{M}$ of size $M$.
5:     Split $\mathcal{M}$ into each GPU (local minibatch $\mathcal{M}_{local}$).
6:     **for** each GPU in parallel **do**
7:         **for** $k = 1, 2, \ldots, K$ **do**
8:             Sample $\epsilon \sim \mathcal{N}(\mathbf{0}, \mathbf{I})$.
9:             $\mathbf{w}^{(k)} \leftarrow \boldsymbol{\mu} + \epsilon\boldsymbol{\sigma}$ with $\boldsymbol{\sigma} \leftarrow (1/(N(\mathbf{s}+\tilde{\delta}+\gamma)))^{1/2}$.
10:            Compute $\mathbf{g}_i^{(k)} \leftarrow \nabla_w \ell(\mathbf{y}_i, \mathbf{f}_{w^{(k)}}(\mathbf{x}_i)), \forall i \in \mathcal{M}_{local}$
             using the method described in Appendix B.
11:            $\hat{\mathbf{g}}_k \leftarrow \frac{1}{M}\sum_{i\in\mathcal{M}_{local}}\mathbf{g}_i^{(k)}$.
12:            $\hat{\mathbf{h}}_k \leftarrow \frac{1}{M}\sum_{i\in\mathcal{M}_{local}}(\mathbf{g}_i^{(k)})^2$.
13:         **end for**
14:         $\hat{\mathbf{g}} \leftarrow \frac{1}{K}\sum_{k=1}^K \hat{\mathbf{g}}_k$ and $\hat{\mathbf{h}} \leftarrow \frac{1}{K}\sum_{k=1}^K \hat{\mathbf{h}}_k$.
15:     **end for**
16:     AllReduce $\hat{\mathbf{g}}, \hat{\mathbf{h}}$.
17:     $\mathbf{m} \leftarrow \beta_1\mathbf{m} + (\hat{\mathbf{g}} + \tilde{\delta}\boldsymbol{\mu})$.
18:     $\mathbf{s} \leftarrow (1 - \tau\beta_2)\mathbf{s} + \beta_2\hat{\mathbf{h}}$.
19:     $\boldsymbol{\mu} \leftarrow \boldsymbol{\mu} - \alpha\mathbf{m}/(\mathbf{s}+\tilde{\delta}+\gamma)$.
20: **until** stopping criterion is met

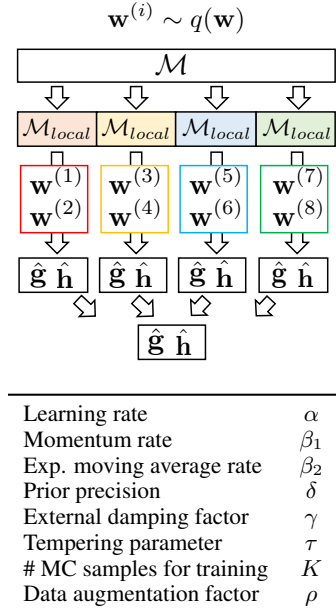

| | |
|---|---|
| Learning rate | $\alpha$ |
| Momentum rate | $\beta_1$ |
| Exp. moving average rate | $\beta_2$ |
| Prior precision | $\delta$ |
| External damping factor | $\gamma$ |
| Tempering parameter | $\tau$ |
| # MC samples for training | $K$ |
| Data augmentation factor | $\rho$ |

Figure 2: A pseudo-code for our distributed VOGN algorithm is shown in Algorithm 1, and the distributed scheme is shown in the right figure. The computation in line 10 requires an extra calculation (see Appendix B), making VOGN slower than Adam. The bottom table gives a list of algorithmic hyperparameters needed for VOGN.

**Momentum and initialisation:** It is well known that both momentum and good initialisation can improve the speed of convergence for SG methods in deep learning [51]. Since VOGN is similar to Adam, we can implement momentum in a similar way. This is shown in step 17 of Algorithm 1, where $\beta_1$ is the momentum rate. We initialise the mean $\boldsymbol{\mu}$ in the same way the weights are initialised in Adam (we use `init.xavier_normal` in PyTorch [11]). For the momentum term $\mathbf{m}$, we use the same initialisation as Adam (initialised to 0). VOGN requires an additional initialisation for the variance $\boldsymbol{\sigma}^2$. For this, we first run a forward pass through the first minibatch, calculate the average of the squared gradients and initialise the scale $\mathbf{s}_0$ with it (see step 1 in Algorithm 1). This implies that the variance is initialised to $\boldsymbol{\sigma}_0^2 = \tau/(N(\mathbf{s}_0 + \tilde{\delta}))$. For the tempering parameter $\tau$, we use a schedule where it is increased from a small value (e.g., 0.1) to 1. With these initialisation protocols, VOGN is able to mimic the convergence behaviour of Adam in the beginning.

**Learning rate scheduling:** A common approach to quickly achieve high validation accuracies is to use a specific learning rate schedule [13]. The learning rate (denoted by $\alpha$ in Algorithm 1) is regularly decayed by a factor (typically a factor of 10). The frequency and timings of this decay are usually pre-specified. In VOGN, we use the same schedule used for Adam, which works well.

**Distributed training:** We also employ distributed training for VOGN to perform large experiments quickly. We can parallelise computation both over data and Monte-Carlo (MC) samples. Data parallelism is useful to split up large minibatch sizes. This is followed by averaging over multiple MC samples and their losses on a single GPU. MC sample parallelism is useful when minibatch size is small, and we can copy the entire minibatch and process it on a single GPU. Algorithm 1 and Figure 2 illustrate our distributed scheme. We use a combination of these two parallelism techniques with different MC samples for different inputs. This theoretically reduces the variance during training (see Equation 5 in Kingma et al. [26]), but sometimes requires averaging over multiple MC samples to get a sufficiently low variance in the early iterations. Overall, we find that this type of distributed training is essential for fast training on large problems such as ImageNet.

**Implementation of the Gauss-Newton update in VOGN:** As discussed earlier, VOGN uses the Gauss-Newton approximation, which is fundamentally different from Adam. In this approximation, the gradients on individual data examples are first squared and then averaged afterwards (see step

12 in Algorithm 1 which implements the update for $\mathbf{s}_t$ shown in (4)). We need extra computation to get access to individual gradients, due to which, VOGN is slower Adam or SGD (e.g., in Fig. 1). However, this is not a theoretical limitation and this can be improved if a framework enables an easy computation of the individual gradients. Details of our implementation are described in Appendix B. This implementation is much more efficient than a naive one where gradients over examples are stored and the sum over the square is computed sequentially. Our implementation usually brings the running time of VOGN to within 2-5 times of the time that Adam takes.

**Tuning VOGN:** Currently, there is no common recipe for tuning the algorithmic hyperparameters for VI, especially for large-scale tasks like ImageNet classification. One key idea we use in our experiments is to start with Adam hyperparameters and then make sure that VOGN training closely follows an Adam-like trajectory in the beginning of training. To achieve this, we divide the tuning into an *optimisation part* and a *regularisation part*. In the *optimisation part*, we first tune the hyperparameters of a deterministic version of VOGN, called the online Gauss-Newton (OGN) method. This method, described in Appendix C, is more stable than VOGN since it does not require MC sampling, and can be used as a stepping stone when moving from Adam/SGD to VOGN. After reaching a competitive performance to Adam/SGD by OGN, we move to the *regularisation part*, where we tune the prior precision $\delta$, the tempering parameter $\tau$, and the number of MC samples $K$ for VOGN. We initialise our search by setting the prior precision $\delta$ using the L2-regularisation parameter used for OGN, as well as the dataset size $N$. Another technique is to warm-up the parameter $\tau$ towards $\tau = 1$ (also see the "momentum and initialisation" part). Setting $\tau$ to smaller values usually stabilises the training, and increasing it slowly also helps during tuning. We also add an *external damping factor* $\gamma > 0$ to the moving average $\mathbf{s}_t$. This increases the lower bound of the eigenvalues of the diagonal covariance $\mathbf{\Sigma}_t$ and prevents the noise and the step size from becoming too large. We find that a mix of these techniques works well for the problems we considered.

## 4    Experiments

In this section, we present experiments on fitting several deep networks on CIFAR-10 and ImageNet. Our experiments demonstrate practical training using VOGN on these benchmarks and show performance that is competitive with Adam and SGD. We also assess the quality of the posterior approximation, finding that benefits of Bayesian principles are preserved.

CIFAR-10 [28] contains 10 classes with 50,000 images for training and 10,000 images for validation. For ImageNet, we train with 1.28 million training examples and validate on 50,000 examples, classifying between 1,000 classes. We used a large minibatch size $M = 4,096$ and parallelise them across 128 GPUs (NVIDIA Tesla P100). We compare the following methods on CIFAR-10: Adam, MC-dropout [9]. For ImageNet, we also compare to SGD, K-FAC, and Noisy K-FAC. We do not consider Noisy K-FAC for other comparisons since tuning is difficult. We compare 3 architectures: LeNet-5, AlexNet, ResNet-18. We only compare to Bayes by Backprop (BBB) [4] for CIFAR-10 with LeNet-5 since it is very slow to converge for larger-scale experiments. We carefully set the hyperparameters of all methods, following the best practice of large distributed training [13] as the initial point of our hyperparameter tuning. The full set of hyperparameters is in Appendix D.

### 4.1   Performance on CIFAR-10 and ImageNet

We start by showing the effectiveness of momentum and batch normalisation for boosting the performance of VOGN. Figure 3a shows that these methods significantly speed up convergence and performance (in terms of both accuracy and log likelihoods).

Figures 1 and 4 compare the convergence of VOGN to Adam (for all experiments), SGD (on ImageNet), and MC-dropout (on the rest). VOGN shows similar convergence and its performance is competitive with these methods. We also try BBB on LeNet-5, where it converges prohibitively slowly, performing very poorly. We are not able to successfully train other architectures using this approach. We found it far simpler to tune VOGN because we can borrow all the techniques used for Adam. Figure 4 also shows the importance of DA in improving performance.

Table 1 gives a final comparison of train/validation accuracies, negative log likelihoods, epochs required for convergence, and run-time per epoch. We can see that the accuracy, log likelihoods, and the number of epochs are comparable. VOGN is 2-5 times slower than Adam and SGD. This

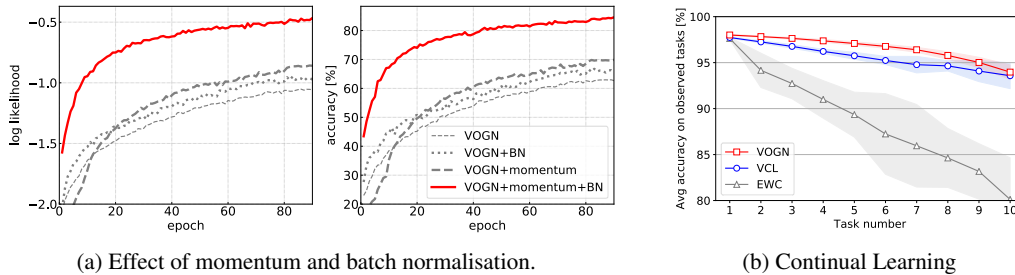

(a) Effect of momentum and batch normalisation.

(b) Continual Learning

Figure 3: Figure (a) shows that momentum and batch normalisation improve the performance of VOGN. The results are for training ResNet-18 on CIFAR-10. Figure (b) shows comparison for a continual-learning task on the Permuted MNIST dataset. VOGN performs at least as well (average accuracy) as VCL over 10 tasks. We also find that, for each task, VOGN converges much faster, taking only 100 epochs per task as opposed to 800 epochs taken by VCL (plots not shown).

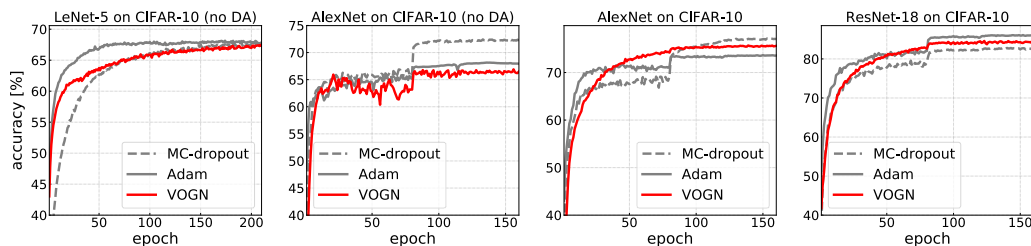

Figure 4: Validation accuracy for various architectures trained on CIFAR-10 (DA: Data Augmentation). VOGN's convergence and validation accuracies are comparable to Adam and MC-dropout.

is mainly due to the computation of individual gradients required in VOGN (see the discussion in Section 3). We clearly see that by using deep-learning techniques on VOGN, we can perform practical deep learning. This is not possible with methods such as BBB.

Due to the Bayesian nature of VOGN, there are some trade-offs to consider. Reducing the prior precision ($\delta$ in Algorithm 1) results in higher validation accuracy, but also larger train-test gap (more overfitting). This is shown in Appendix E for VOGN on ResNet-18 on ImageNet. As expected, when the prior precision is small, performance is similar to non-Bayesian methods. We also show the effect of changing the effective dataset size $\rho$ in Appendix E: note that, since we are going to tune the prior variance $\delta$ anyway, it is sufficient to set $\rho$ to its correct order of magnitude. Another trade-off concerns the number of Monte-Carlo (MC) samples, shown in Appendix F. Increasing the number of training MC samples (up to a limit) improves VOGN's convergence rate and stability, but also increases the computation. Increasing the number of MC samples during testing improves generalisation, as expected due to averaging.

Finally, a few comments on the performance of the other methods. Adam regularly overfits the training set in most settings, with large train-test differences in both validation accuracy and log likelihood. One exception is LeNet-5, which is most likely due to the small architecture which results in underfitting (this is consistent with the low validation accuracies obtained). In contrast to Adam, MC-dropout has small train-test gap, usually smaller than VOGN's. However, we will see in Section 4.2 that this is because of underfitting. Moreover, the performance of MC-dropout is highly sensitive to the dropout rate (see Appendix G for a comparison of different dropout rates). On ImageNet, Noisy K-FAC performs well too. It is slower than VOGN, but it takes fewer epochs. Overall, wall clock time is about the same as VOGN.

## 4.2 Quality of the Predictive Probabilities

In this section, we compare the quality of the predictive probabilities for various methods. For Bayesian methods, we compute these probabilities by averaging over the samples from the posterior approximations (see Appendix H for details). For non-Bayesian methods, these are obtained using the

| Dataset/ Architecture | Optimiser | Train/Validation Accuracy (%) | Validation NLL | Epochs | Time/ epoch (s) | ECE | AUROC |
|---|---|---|---|---|---|---|---|
| CIFAR-10/ LeNet-5 (no DA) | Adam | 71.98 / **67.67** | **0.937** | 210 | 6.96 | **0.021** | 0.794 |
| | BBB | 66.84 / 64.61 | 1.018 | 800 | 11.43$^\dagger$ | 0.045 | 0.784 |
| | MC-dropout | 68.41 / **67.65** | 0.99 | 210 | 6.95 | 0.087 | **0.797** |
| | VOGN | 70.79 / **67.32** | **0.938** | 210 | 18.33 | 0.046 | **0.8** |
| CIFAR-10/ AlexNet (no DA) | Adam | 100.0 / 67.94 | 2.83 | 161 | 3.12 | 0.262 | 0.793 |
| | MC-dropout | 97.56 / **72.20** | 1.077 | 160 | 3.25 | 0.140 | **0.818** |
| | VOGN | 79.07 / 69.03 | **0.93** | 160 | 9.98 | **0.024** | 0.796 |
| CIFAR-10/ AlexNet | Adam | 97.92 / 73.59 | 1.480 | 161 | 3.08 | 0.262 | 0.793 |
| | MC-dropout | 80.65 / **77.04** | **0.667** | 160 | 3.20 | 0.114 | 0.828 |
| | VOGN | 81.15 / 75.48 | 0.703 | 160 | 10.02 | **0.016** | **0.832** |
| CIFAR-10/ ResNet-18 | Adam | 97.74 / **86.00** | 0.55 | 160 | 11.97 | 0.082 | **0.877** |
| | MC-dropout | 88.23 / 82.85 | 0.51 | 161 | 12.51 | 0.166 | 0.768 |
| | VOGN | 91.62 / 84.27 | **0.477** | 161 | 53.14 | **0.040** | **0.876** |
| ImageNet/ ResNet-18 | SGD | 82.63 / **67.79** | **1.38** | 90 | 44.13 | 0.067 | 0.856 |
| | Adam | 80.96 / 66.39 | 1.44 | 90 | 44.40 | 0.064 | 0.855 |
| | MC-dropout | 72.96 / 65.64 | 1.43 | 90 | 45.86 | **0.012** | 0.856 |
| | OGN | 85.33 / 65.76 | 1.60 | 90 | 63.13 | 0.128 | 0.854 |
| | VOGN | 73.87 / **67.38** | **1.37** | 90 | 76.04 | 0.029 | 0.854 |
| | K-FAC | 83.73 / 66.58 | 1.493 | 60 | 133.69 | 0.158 | 0.842 |
| | Noisy K-FAC | 72.28 / 66.44 | 1.44 | 60 | 179.27 | 0.080 | 0.852 |

Table 1: Performance comparisons on different dataset/architecture combinations. Out of the 15 metrics (NLL, ECE, and AUROC on 5 dataset/architecture combinations), VOGN performs the best or tied best on 10 ,and is second-best on the other 5. Here DA means 'Data Augmentation', NLL refers to 'Negative Log Likelihood' (lower is better), ECE refers to 'Expected Calibration Error' (lower is better), AUROC refers to 'Area Under ROC curve' (higher is better). BBB is the Bayes By Backprop method. For ImageNet, the reported accuracy and negative log likelihood are the median value from the final 5 epochs. All hyperparameter settings are in Appendix D. See Table 3 for standard deviations. $^\dagger$ BBB is not parallelised (other methods have 4 processes), with 1 MC sample used for the convolutional layers (VOGN uses 6 samples per process).

point estimate of the weights. We compare the probabilities using the following metrics: validation negative log-likelihood (NLL), area under ROC (AUROC) and expected calibration curves (ECE) [40, 15]. For the first and third metric, a lower number is better, while for the second, a higher number is better. See Appendix H for an explanation of these metrics. Results are summarised in Table 1. VOGN's uncertainty performance is more consistent and marginally better than the other methods, as expected from a more principled Bayesian method. Out of the 15 metrics (NLL, ECE and AUROC on 5 dataset/architecture combinations), VOGN performs the best or tied best on 10, and is second-best on the other 5. In contrast, both MC-dropout's and Adam's performance varies significantly, sometimes performing poorly, sometimes performing decently. MC-dropout is best on 4, and Adam is best on 1 (on LeNet-5; as argued earlier, the small architecture may result in underfitting). We also show calibration curves [7] in Figures 1 and 14. Adam is consistently over-confident, with its calibration curve below the diagonal. Conversely, MC-dropout is usually under-confident. On ImageNet, MC-dropout performs well on ECE (all methods are very similar on AUROC), but this required an excessively tuned dropout rate (see Appendix G).

We also compare performance on out-of-distribution datasets. When testing on datasets that are different from the training datasets, predictions should be more uncertain. We use experimental protocol from the literature [16, 31, 8, 32] to compare VOGN, Adam and MC-dropout on CIFAR-10. We also borrow metrics from other works [16, 30], showing predictive entropy histograms and also reporting AUROC and FPR at 95% TPR. See Appendix I for further details on the datasets and metrics. Ideally, we want predictive entropy to be high on out-of-distribution data and low on in-distribution data. Our results are summarised in Figure 5 and Appendix I. On ResNet-18 and AlexNet, VOGN's predictive entropy histograms show the desired behaviour: a spread of entropies for the in-distribution data, and high entropies for out-of-distribution data. Adam has many predictive entropies at zero,

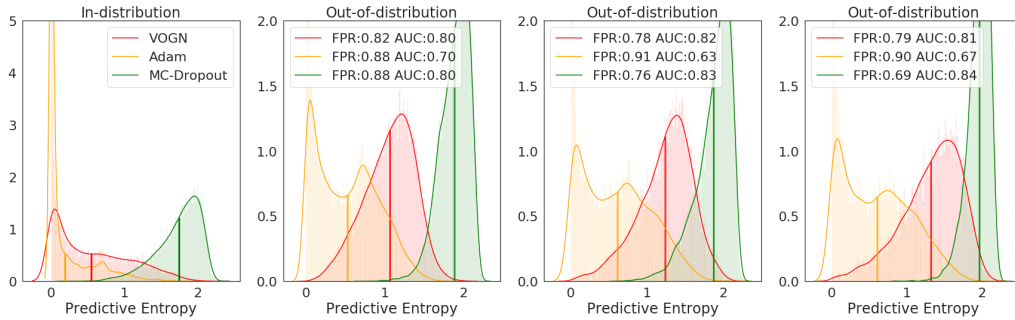

Figure 5: Histograms of predictive entropy for out-of-distribution tests for ResNet-18 trained on CIFAR-10. Going from left to right, the inputs are: the in-distribution dataset (CIFAR-10), followed by out-of-distribution data: SVHN, LSUN (crop), LSUN (resize). Also shown are the FPR at 95% TPR metric (lower is better) and the AUROC metric (higher is better), averaged over 3 runs. We clearly see that VOGN's predictive entropy is generally low for in-distribution and high for out-of-distribution data, but this is not the case for other methods. Solid vertical lines indicate the mean predictive entropy. The standard deviations are small and therefore not reported.

indicating Adam tends to classify out-of-distribution data too confidently. Conversely, MC-dropout's predictive entropies are generally high (particularly in-distribution), indicating MC-dropout has too much noise. On LeNet-5, we observe the same result as before: Adam and MC-dropout both perform well. The metrics (AUROC and FPR at 95% TPR) do not provide a clear story across architectures.

### 4.2.1 Performance on a Continual-learning task

The goal of continual learning is to avoid forgetting of old tasks while sequentially observing new tasks. The past tasks are never visited again, making it difficult to remember them. The field of continual learning has recently grown, with many approaches proposed to tackle this problem [27, 33, 43, 48, 50]. Most approaches consider a simple setting where the tasks (such as classifying a subset of classes) arrive sequentially, and all the data from that task is available. We consider the same setup in our experiments.

We compare to Elastic Weight Consolidation (EWC) [27] and a VI-based approach called Variational Continual Learning (VCL) [43]. VCL employs BBB for each task, and we expect to boost its performance by replacing BBB by VOGN. Figure 3b shows results on a common benchmark called Permuted MNIST. We use the same experimental setup as in Swaroop et al. [52]. In Permuted MNIST, each task consists of the entire MNIST dataset (10-way classification) with a different fixed random permutation applied to the input images' pixels. We run each method 20 times, with different random seeds for both the benchmark's permutations and model training. See Appendix D.2 for hyperparameter settings and further details. We see that VOGN performs at least as well as VCL, and far better than a popular approach called EWC [27]. Additionally, as found in the batch learning setting, VOGN is much quicker than BBB: we run VOGN for only 100 epochs per task, whereas VCL requires 800 epochs per task to achieve best results [52].

## 5 Conclusions

We successfully train deep networks with a natural-gradient variational inference method, VOGN, on a variety of architectures and datasets, even scaling up to ImageNet. This is made possible due to the similarity of VOGN to Adam, enabling us to boost performance by borrowing deep-learning techniques. Our accuracies and convergence rates are comparable to SGD and Adam. Unlike them, however, VOGN retains the benefits of Bayesian principles, with well-calibrated uncertainty and good performance on out-of-distribution data. Better uncertainty estimates open up a whole range of potential future experiments, for example, small data experiments, active learning, adversarial experiments, and sequential decision making. Our results on a continual-learning task confirm this. Another potential avenue for research is to consider structured covariance approximations.

**Acknowledgements**

We would like to thank Hikaru Nakata (Tokyo Institute of Technology) and Ikuro Sato (Denso IT Laboratory, Inc.) for their help on the PyTorch implementation. We are also thankful for the RAIDEN computing system and its support team at the RIKEN Center for AI Project which we used extensively for our experiments. This research used computational resources of the HPCI system provided by Tokyo Institute of Technology (TSUBAME3.0) through the HPCI System Research Project (Project ID:hp190122). K. O. is a Research Fellow of JSPS and is supported by JSPS KAKENHI Grant Number JP19J13477.

## Footnotes

* These two authors contributed equally.

† This work is conducted during an internship at RIKEN Center for AI project.

‡ Corresponding author: `emtiyaz.khan@riken.jp`

[1] The code is available at `https://github.com/team-approx-bayes/dl-with-bayes`.

[2]This regulariser is sometimes set to 0 or a very small value.

[3]Alternate versions with weight-decay and momentum differ from this update [34]. We present a form useful to establish the connection between SG methods and natural-gradient VI.

[4]This is a tempered posterior [54] setup where $\tau$ is set $\neq 1$ when we expect model misspecification and/or adversarial examples [10]. Setting $\tau = 1$ recovers standard Bayesian inference.

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
