[Supplementary Material · supplement.pdf]

## A  Noisy K-FAC algorithm

Noisy K-FAC [56] attempts to approximate the structure of the full covariance matrix, and therefore the updates are a bit more involved than VOGN (see Equation 4). Assuming a fully-connected layer, we denote the weight matrix of layer by $\mathbf{W}$. The Noisy K-FAC method estimates the parameters of a matrix-variate Gaussian distribution $q_t(\mathbf{W}) = \mathcal{MN}(\mathbf{W}|\mathbf{M}_t, \boldsymbol{\Sigma}_{2,t} \otimes \boldsymbol{\Sigma}_{1,t})$ by using the following updates:

$$\mathbf{M}_{t+1} \leftarrow \mathbf{M}_t - \alpha \left[\mathbf{A}_{t+1}^{\gamma}\right]^{-1} \left(\nabla_W E\left[\ell(y_i, f_W(\mathbf{x}_i))\right] + \tilde{\delta}\mathbf{W}_t\right) \left[\mathbf{S}_{t+1}^{\gamma}\right]^{-1}, \tag{5}$$

$$\mathbf{A}_{t+1} \leftarrow (1 - \tilde{\beta}_t)\mathbf{A}_t + \tilde{\beta}_t E\left[\mathbf{a}_t \mathbf{a}_t^{\top}\right], \quad \mathbf{S}_{t+1} \leftarrow (1 - \tilde{\beta}_t)\mathbf{S}_t + \tilde{\beta}_t E\left[\mathbf{g}_t \mathbf{g}_t^{\top}\right], \tag{6}$$

where $\mathbf{W}_t \sim q_t(\mathbf{W})$, $\mathbf{g}_t := \nabla_s \ell(y_i, f_W(\mathbf{x}_i))$ with $s = s_t := \mathbf{W}_t^{\top}\mathbf{a}_t$, $\mathbf{a}_t$ is the input vector (the activation of the previous layer), $E\left[\cdot\right]$ is the average over the minibatch. $\tilde{\beta} := \beta\tau/N$, and $\gamma := \tilde{\gamma} + \gamma_{ex}$ with some *external* damping factor $\gamma_{ex}$. The covariance parameters are set to $\boldsymbol{\Sigma}_{2,t}^{-1} := \tau\mathbf{A}_t^{\gamma}/N$ and $\boldsymbol{\Sigma}_{1,t}^{-1} := \mathbf{S}_t^{\gamma}$, where $\mathbf{A}_t^{\gamma} := \mathbf{A}_t + \pi_t\sqrt{\gamma}\mathbf{I}$ and $\mathbf{S}_t^{\gamma} := \mathbf{S}_t + \frac{1}{\pi_t}\sqrt{\gamma}\mathbf{I}$. $\pi_t^2(\pi_t > 0)$ is the average eigenvalue of $\mathbf{A}_t$ divided by that of $\mathbf{S}_t$. Similarly to the VOGN update in Equation 4, the gradients are scaled by matrices $\mathbf{A}_t$ and $\mathbf{S}_t$, which are related to the precision matrix of the approximation.

## B  Details on fast implementation of the Gauss-Newton approximation

Current codebases are only optimised to directly return the average of gradients over the minibatch. In order to efficiently compute the Gauss-Newton (GN) approximation, we modify the backward-pass to efficiently calculate the gradient per example in the minibatch, and extend the solution in Goodfellow [12] to both convolutional and batch normalisation layers.

### B.1  Convolutional layer

Consider a convolutional layer with a weight matrix $\mathbf{W} \in \mathbb{R}^{C_{out} \times C_{in} k^2}$ (ignore bias for simplicity) and an input tensor $\mathbf{A} \in \mathbb{R}^{C_{in} \times H_{in} \times W_{in}}$, where $C_{out}, C_{in}$ are the number of output, input channels, respectively, $H_{in}, W_{in}$ are the spatial dimensions, and $k$ is the kernel size. For any stride and padding, by applying `torch.nn.functional.unfold` function in PyTorch[5], we get the extended matrix $\mathbf{M}_A \in \mathbb{R}^{C_{in} k^2 \times H_{out} W_{out}}$ so that the output tensor $\mathbf{S}$ is calculated by a matrix multiplication:

$$\mathbf{M}_A \leftarrow \text{unfold}\left(\mathbf{A}\right) \in \mathbb{R}^{C_{in} k^2 \times H_{out} W_{out}}, \tag{7}$$

$$\mathbf{M}_S \leftarrow \mathbf{W}\mathbf{M}_A \in \mathbb{R}^{C_{out} \times H_{out} W_{out}}, \tag{8}$$

$$\mathbf{S} \leftarrow \text{reshape}\left(\mathbf{M}_S\right) \in \mathbb{R}^{C_{out} \times H_{out} \times W_{out}}, \tag{9}$$

where $H_{out}, W_{out}$ are the spatial dimensions of the output feature map. Using the matrix $\mathbf{M}_A$, we can also get the gradient per example by a matrix multiplication:

$$\nabla_{M_S} \ell(y_i, f_W(\mathbf{x}_i)) \leftarrow \text{reshape}\left(\nabla_S \ell(y_i, f_W(\mathbf{x}_i))\right) \in \mathbb{R}^{C_{out} \times H_{out} W_{out}}, \tag{10}$$

$$\nabla_W \ell(y_i, f_W(\mathbf{x}_i)) \leftarrow \nabla_{M_S} \ell(y_i, f_W(\mathbf{x}_i))\mathbf{M}_A^{\top} \in \mathbb{R}^{C_{out} \times C_{in} k^2}. \tag{11}$$

Note that in PyTorch, we can access to the inputs $\mathbf{A}$ and the gradient $\nabla_S \ell(y_i, f_W(\mathbf{x}_i))$ per example in the computational graph during a forward-pass and a backward-pass, respectively, by using the `Function Hooks` [6]. Hence, to get the gradient $\nabla_W \ell(y_i, f_W(\mathbf{x}_i))$ per example, we only need to perform (7), (10), and (11) after the backward-pass for this layer.

### B.2  Batch normalisation layer

Consider a batch normalisation layer follows a fully-connected layer, which activation is $\mathbf{a} \in \mathbb{R}^d$, with the scale parameter $\boldsymbol{\gamma} \in \mathbb{R}^d$ and the shift parameter $\boldsymbol{\beta} \in \mathbb{R}^d$, we get the output of this layer

$$\mathbf{H} = E\left[\nabla \log p(y|\boldsymbol{x}) \nabla \log p(y|\boldsymbol{x})^{\mathrm{T}}\right]$$

Figure 6: Layer-wise block-diagonal Gauss-Newton approximation

$\mathbf{s} \in \mathbb{R}^d$ by,

$$\boldsymbol{\mu} \leftarrow E\left[\mathbf{a}\right] \in \mathbb{R}^d, \tag{12}$$

$$\boldsymbol{\sigma}^2 \leftarrow E\left[(\mathbf{a} - \boldsymbol{\mu})^2\right] \in \mathbb{R}^d, \tag{13}$$

$$\hat{\mathbf{a}} \leftarrow \frac{\mathbf{a} - \boldsymbol{\mu}}{\sqrt{\boldsymbol{\sigma}^2}} \in \mathbb{R}^d, \tag{14}$$

$$\mathbf{s} \leftarrow \boldsymbol{\gamma}\hat{\mathbf{a}} + \boldsymbol{\beta} \in \mathbb{R}^d, \tag{15}$$

where $E\left[\cdot\right]$ is the average over the minibatch and $\hat{\mathbf{a}}$ is the normalised input. We can find the gradient with respect to parameters $\boldsymbol{\gamma}$ and $\boldsymbol{\beta}$ per example by,

$$\nabla_\gamma \ell(y_i, f_W(\mathbf{x}_i) \leftarrow \nabla_s \ell(y_i, f_W(\mathbf{x}_i) \circ \hat{\mathbf{a}}, \tag{16}$$

$$\nabla_\beta \ell(y_i, f_W(\mathbf{x}_i) \leftarrow \nabla_s \ell(y_i, f_W(\mathbf{x}_i). \tag{17}$$

We can obtain the input $\mathbf{a}$ and the gradient $\nabla_s \ell(y_i, f_W(\mathbf{x}_i)$ per example from the computational graph in PyTorch in the same way as a convolutional layer.

### B.3  Layer-wise block-diagonal Gauss-Newton approximation

Despite using the method above, it is still intractable to compute the Gauss-Newton matrix (and its inverse) with respect to the weights of large-scale deep neural networks. We therefore apply two further approximations (Figure 6). First, we view the Gauss-Newton matrix as a layer-wise block-diagonal matrix. This corresponds to ignoring the correlation between the weights of different layers. Hence for a network with $L$ layers, there are $L$ diagonal blocks, and $\mathbf{H}_\ell$ is the diagonal block corresponding to the $\ell$-th layer ($\ell = 1, \ldots, L$). Second, we approximate each diagonal block $\mathbf{H}_\ell$ with $\tilde{\mathbf{H}}_\ell$, which is either a Kronecker-factored or diagonal matrix. Using a Kronecker-factored matrix as $\tilde{\mathbf{H}}_\ell$ corresponds to K-FAC; a diagonal matrix corresponds to a mean-field approximation in that layer. By applying these two approximations, the update rule of the Gauss-Newton method can be written in a layer-wise fashion:

$$\boldsymbol{W}_{\ell,t+1} = \boldsymbol{W}_{\ell,t} - \alpha_t \tilde{\mathbf{H}}_\ell(\boldsymbol{\theta}_t)^{-1} \mathbf{g}_\ell(\boldsymbol{\theta}_t) \ (\ell = 1, \ldots, L), \tag{18}$$

where $\boldsymbol{W}_\ell$ is the weights in $\ell$-th layer, and

$$\boldsymbol{\theta} = \left( \ \mathrm{vec}(\boldsymbol{W}_1)^{\mathrm{T}} \quad \cdots \quad \mathrm{vec}(\boldsymbol{W}_\ell)^{\mathrm{T}} \quad \cdots \quad \mathrm{vec}(\boldsymbol{W}_{\mathrm{L}})^{\mathrm{T}} \ \right)^{\mathrm{T}}. \tag{19}$$

Since the cost of computing $\tilde{\mathbf{H}}_\ell^{-1}$ is much cheaper compared to that of computing $\mathbf{H}^{-1}$, our approximations make Gauss-Newton much more practical in deep learning.

In the distributed setting (see Figure 2), each parallel process (corresponding to 1 GPU) calculates the GN matrix for its local minibatch. Then, one GPU adds them together and calculates the inverse. This inversion step can also be parallelised after making the block-diagonal approximation to the GN matrix. After inverting the GN matrix, the standard deviation $\boldsymbol{\sigma}$ is updated (line 9 in Algorithm 1), and sent to each parallel process, allowing each process to draw independently from the posterior.

In the Noisy K-FAC case, a similar distributed scheme is used, except each parallel process now has both matrices $\mathbf{S}$ and $\mathbf{A}$ (see Appendix A). When using K-FAC approximations to the Gauss-Newton blocks for other layers, Osawa et al. [44] empirically showed that the BatchNorm layer can be approximated with a diagonal matrix without loss of accuracy, and we find the same. We therefore use diagonal $\tilde{\mathbf{H}}_\ell$ with K-FAC and Noisy K-FAC in BatchNorm layers (see Table 2). For further details on how to efficiently parallelise K-FAC in the distributed setting, please see Osawa et al. [44].

| optimiser | convolution | fully-connected | Batch Normalisation |
|---|---|---|---|
| OGN | diagonal | diagonal | diagonal |
| VOGN | diagonal | diagonal | diagonal |
| K-FAC | Kronecker-factored | Kronecker-factored | diagonal |
| Noisy K-FAC | Kronecker-factored | Kronecker-factored | diagonal |

Table 2: The approximation used for each layer type's diagonal block $\tilde{\mathbf{H}}_\ell$ for the different optimisers tested this paper.

## C  OGN: A deterministic version of VOGN

To easily apply the tricks and techniques of deep-learning methods, we recommend to first test them on a deterministic version of VOGN, which we call the online Gauss-Newton (OGN) method. In this method, we approximate the gradients at the mean of the Gaussian, rather than using MC samples[7]. This results in an update without any sampling, as shown below (we have replaced $\boldsymbol{\mu}_t$ by $\mathbf{w}_t$ since there is no distinction between them):

$$\mathbf{w}_{t+1} \leftarrow \mathbf{w}_t - \alpha_t \frac{\hat{\mathbf{g}}(\mathbf{w}_t) + \tilde{\delta}\mathbf{w}_t}{\mathbf{s}_{t+1} + \tilde{\delta}}, \quad \mathbf{s}_{t+1} \leftarrow (1 - \tau\beta_t)\mathbf{s}_t + \beta_t \frac{1}{M} \sum_{i \in \mathcal{M}_t} \left(\mathbf{g}_i(\mathbf{w}_t)\right)^2. \quad (20)$$

At each iteration, we still get a Gaussian $\mathcal{N}(\mathbf{w}|\mathbf{w}_t, \hat{\boldsymbol{\Sigma}}_t)$ with $\hat{\boldsymbol{\Sigma}}_t := \mathrm{diag}(1/(N(\mathbf{s}_t + \tilde{\delta})))$. It is easy to see that, like SG methods, this algorithm converges to a local minima of the loss function, thereby obtaining a Laplace approximation instead of a variational approximation. The advantage of OGN is that this can be used as a stepping stone, when switching from Adam to VOGN. Since it does not involve sampling, the tricks and techniques applied to Adam are easier to apply to OGN than VOGN. However, due to the lack of averaging over samples, this algorithm is less effective to preserve the benefits of Bayesian principles, and gives slightly worse uncertainty estimates.

## D  Hyperparameter settings

Hyperparameters for all results shown in Table 1 are given in Table 4. The settings for distributed VI training are given in Table 5. Please see Goyal et al. [13] and Osawa et al. [44] for best practice on these hyperparameter values.

### D.1  Bayes by Backprop for CIFAR-10/LeNet-5 training

We use hyperparameter settings and training procedure for Bayes by Backprop (BBB) [4] as suggested by Swaroop et al. [52]. This includes using the local reparameterisation trick, initialising means and variances at small values, using 10 MC samples per minibatch during training for linear layers (1 MC sample per minibatch for convolutional layers) and 100 MC samples per minibatch during testing for linear layers (10 MC samples per minibatch for convolutional layers). Note that BBB has twice as many parameters to optimise than Adam or SGD (means and variances for each weight in the deep neural network). The fewer MC samples per minibatch for convolutional layers speed up training time per epoch while empirically not reducing convergence rate.

| Dataset/ Architecture | Optimiser | Train Acc (%) | Train NLL | Validation Acc (%) | Validation NLL | ECE | AUROC | Epochs | Time/ epoch (s) |
|---|---|---|---|---|---|---|---|---|---|
| CIFAR-10/ LeNet-5 (no DA) | Adam | 71.98 ± 0.117 | 0.733 ± 0.021 | **67.67 ± 0.513** | **0.937 ± 0.012** | **0.021 ± 0.002** | 0.794 ± 0.001 | 210 | 6.96 |
| | BBB | 66.84 ± 0.003 | 0.957 ± 0.006 | 64.61 ± 0.331 | 1.018 ± 0.006 | 0.045 ± 0.005 | 0.784 ± 0.003 | 800 | 11.43 |
| | MC-dropout | 68.41 ± 0.581 | 0.870 ± 0.101 | **67.65 ± 1.317** | 0.99 ± 0.026 | 0.087 ± 0.009 | **0.797 ± 0.006** | 210 | 6.95 |
| | VOGN | 70.79 ± 0.763 | 0.880 ± 0.02 | 67.32 ± 1.310 | **0.938 ± 0.024** | 0.046 ± 0.002 | **0.8 ± 0.002** | 210 | 18.33 |
| CIFAR-10/ AlexNet (no DA) | Adam | 100.0 ± 0 | 0.001 ± 0 | 67.94 ± 0.537 | 2.83 ± 0.02 | 0.262 ± 0.005 | 0.793 ± 0.001 | 161 | 3.12 |
| | MC-dropout | 97.56 ± 0.278 | 0.058 ± 0.014 | **72.20 ± 0.177** | 1.077 ± 0.012 | 0.140 ± 0.004 | **0.818 ± 0.002** | 160 | 3.25 |
| | VOGN | 79.07 ± 0.248 | 0.696 ± 0.020 | 69.03 ± 0.419 | **0.931 ± 0.017** | **0.024 ± 0.010** | 0.796 ± 0 | 160 | 9.98 |
| CIFAR-10/ AlexNet | Adam | 97.92 ± 0.140 | 0.057 ± 0.006 | 73.59 ± 0.296 | 1.480 ± 0.015 | 0.262 ± 0.005 | 0.793 ± 0.001 | 161 | 3.08 |
| | MC-dropout | 80.65 ± 0.615 | 0.47 ± 0.052 | **77.04 ± 0.343** | **0.667 ± 0.012** | 0.114 ± 0.002 | 0.828 ± 0.002 | 160 | 3.20 |
| | VOGN | 81.15 ± 0.259 | 0.511 ± 0.039 | 75.48 ± 0.478 | 0.703 ± 0.006 | **0.016 ± 0.001** | **0.832 ± 0.002** | 160 | 10.02 |
| CIFAR-10/ ResNet-18 | Adam | 97.74 ± 0.140 | 0.059 ± 0.012 | **86.00 ± 0.257** | 0.55 ± 0.01 | 0.082 ± 0.002 | **0.877 ± 0.001** | 160 | 11.97 |
| | MC-dropout | 88.23 ± 0.243 | 0.317 ± 0.045 | 82.85 ± 0.208 | 0.51 ± 0 | 0.166 ± 0.025 | 0.768 ± 0.004 | 161 | 12.51 |
| | VOGN | 91.62 ± 0.07 | 0.263 ± 0.051 | 84.27 ± 0.195 | **0.477 ± 0.006** | **0.040 ± 0.002** | **0.876 ± 0.002** | 161 | 53.14 |
| ImageNet/ ResNet-18 | SGD | 82.63 ± 0.058 | 0.675 ± 0.017 | **67.79 ± 0.017** | **1.38 ± 0** | 0.067 | 0.856 | 90 | 44.13 |
| | Adam | 80.96 ± 0.098 | 0.723 ± 0.015 | 66.39 ± 0.168 | 1.44 ± 0.01 | 0.064 | 0.855 | 90 | 44.40 |
| | MC-dropout | 72.96 | 1.12 | 65.64 | 1.43 | **0.012** | 0.856 | 90 | 45.86 |
| | OGN | 85.33 ± 0.057 | 0.526 ± 0.005 | 65.76 ± 0.115 | 1.60 ± 0.00 | 0.128 ± 0.004 | 0.8543 ± 0.001 | 90 | 63.13 |
| | VOGN | 73.87 ± 0.061 | 1.02 ± 0.01 | **67.38 ± 0.263** | **1.37 ± 0.01** | 0.0293 ± 0.001 | 0.8543 ± 0 | 90 | 76.04 |
| | K-FAC | 83.73 ± 0.058 | 0.571 ± 0.016 | 66.58 ± 0.176 | 1.493 ± 0.006 | 0.158 ± 0.005 | 0.842 ± 0.005 | 60 | 133.69 |
| | Noisy K-FAC | 72.28 | 1.075 | 66.44 | 1.44 | 0.080 | 0.852 | 60 | 179.27 |

Table 3: Comparing optimisers on different dataset/architecture combinations, means and standard deviations over three runs. DA: Data Augmentation, Acc.: Accuracy (higher is better), NLL: Negative Log Likelihood (lower is better), ECE: Expected Calibration Error (lower is better), AUROC: Area Under ROC curve (higher is better), BBB: Bayes By Backprop. For ImageNet results, the reported accuracy and negative log likelihood are the median value from the final 5 epochs.

| Dataset/ Architecture | Optimiser | $\alpha_{init}$ | $\alpha$ | Epochs to decay $\alpha$ | $\beta_1$ | $\beta_2$ | Weight decay | L2 reg |
|---|---|---|---|---|---|---|---|---|
| CIFAR-10/ LeNet-5 (no DA) | Adam | - | 1e-3 | - | 0.1 | 0.001 | 1e-2 | - |
| | BBB | - | 1e-3 | - | 0.1 | 0.001 | - | - |
| | MC-dropout | - | 1e-3 | - | 0.9 | - | - | 1e-4 |
| | VOGN | - | 1e-2 | - | 0.9 | 0.999 | - | - |
| CIFAR-10/ AlexNet (no DA) | Adam | - | 1e-3 | [80, 120] | 0.1 | 0.001 | 1e-4 | - |
| | MC-dropout | - | 1e-1 | [80, 120] | 0.9 | - | - | 1e-4 |
| | VOGN | - | 1e-4 | [80, 120] | 0.9 | 0.999 | - | - |
| CIFAR-10/ AlexNet | Adam | - | 1e-3 | [80, 120] | 0.1 | 0.001 | 1e-4 | - |
| | MC-dropout | - | 1e-1 | [80, 120] | 0.9 | - | - | 1e-4 |
| | VOGN | - | 1e-4 | [80, 120] | 0.9 | 0.999 | - | - |
| CIFAR-10/ ResNet-18 | Adam | - | 1e-3 | [80, 120] | 0.1 | 0.001 | 5e-4 | - |
| | MC-dropout | - | 1e-1 | [80, 120] | 0.9 | - | - | 1e-4 |
| | VOGN | - | 1e-4 | [80, 120] | 0.9 | 0.999 | - | - |
| ImageNet/ ResNet-18 | SGD | 1.25e-2 | 1.6 | [30, 60, 80] | 0.9 | - | - | 1e-4 |
| | Adam | 1.25e-5 | 1.6e-3 | [30, 60, 80] | 0.1 | 0.001 | 1e-4 | - |
| | MC-dropout | 1.25e-2 | 1.6 | [30, 60, 80] | 0.9 | - | - | 1e-4 |
| | OGN | 1.25e-5 | 1.6e-3 | [30, 60, 80] | 0.9 | 0.9 | - | 1e-5 |
| | VOGN | 1.25e-5 | 1.6e-3 | [30, 60, 80] | 0.9 | 0.999 | - | - |
| | K-FAC | 1.25e-5 | 1.6e-3 | [15, 30, 45] | 0.9 | 0.9 | - | 1e-4 |
| | Noisy K-FAC | 1.25e-5 | 1.6e-3 | [15, 30, 45] | 0.9 | 0.9 | - | - |

Table 4: Hyperparameters for all results in Table 1

| Optimiser | Dataset/ Architecture | $M$ | # GPUs | $K$ | $\tau$ | $\rho$ | $N_{orig}$ | $\delta$ | $\tilde{\delta}$ | $\gamma$ |
|---|---|---|---|---|---|---|---|---|---|---|
| VOGN | CIFAR-10/ LeNet-5 (no DA) | 128 | 4 | 6 | $0.1 \to 1$ | 1 | 50,000 | 100 | 2e-4 → 2e-3 | 1e-3 |
| | CIFAR-10/ AlexNet (no DA) | 128 | 8 | 3 | $0.05 \to 1$ | 1 | 50,000 | 0.5 | 5e-7 → 1e-5 | 1e-3 |
| | CIFAR-10/ AlexNet | 128 | 8 | 3 | $0.5 \to 1$ | 10 | 50,000 | 0.5 | 5e-7 → 1e-5 | 1e-3 |
| | CIFAR-10/ ResNet-18 | 256 | 8 | 5 | 1 | 10 | 50,000 | 50 | 1e-3 | 1e-3 |
| | ImageNet/ ResNet-18 | 4096 | 128 | 1 | 1 | 5 | 1,281,167 | 133.3 | 2e-5 | 1e-4 |
| Noisy K-FAC | ImageNet/ ResNet-18 | 4096 | 128 | 1 | 1 | 5 | 1,281,167 | 133.3 | 2e-5 | 1e-4 |

Table 5: Settings for distributed VI training

## D.2 Continual learning experiment

Following the setup of Swaroop et al. [52], all models are run with two hidden layers, of 100 hidden units each, with ReLU activation functions. VCL is run with the same hyperparameter settings as in Swaroop et al. [52]. We perform a grid search over EWC's $\lambda$ hyperparameter, finding that $\lambda = 100$ performs the best, exactly like in Nguyen et al. [43].

VOGN is run for 100 epochs per task. Parameters are initialised before training with the default PyTorch initialisation for linear layers. The initial precision is 1e6. A standard normal initial prior is used, just like in VCL. Between tasks, the mean and precision are initialised in the same way as for the first task. The learning rate $\alpha = 1e - 3$, the batch size $M = 256$, $\beta_1 = 0$, $\beta_2 = 1e - 3$, 10 MC samples are used during training and 100 for testing. We run each method 20 times, with different random seeds for both the benchmark's permutation and for model training.

## E  Effect of prior variance and dataset size reweighting factor

We show the effect of changing the prior variance ($\delta^{-1}$ in Algorithm 1) in Figures 8 and 9. We can see that increasing the prior variance improves validation performance (accuracy and log likelihood). However, increasing prior variance also always increases the train-test gap, without exceptions, when

the other hyperparameters are held constant. As an example, training VOGN on ResNet-18 on ImageNet with a prior variance of $7.5e - 4$ has train-test accuracy and log likelihood gaps of 2.29 and 0.12 respectively. When the prior variance is increased to $7.5e - 3$, the respective train-test gaps increase to 6.38 and 0.34 (validation accuracy and validation log likelihood also increase, see Figure 8).

With increased prior variance, VOGN (and Noisy K-FAC) reach converged solutions more like their non-Bayesian counterparts, where overfitting is an issue. This is as expected from Bayesian principles.

Figure 10 shows the combined effect of the dataset reweighting factor $\rho$ and prior variance. When $\rho$ is set to a value in the correct order of magnitude, it does not affect performance so much: instead, we should tune $\delta$. This is our methodology when dealing with $\rho$. Note that we set $\rho$ for ImageNet to be smaller than that for CIFAR-10 because the data augmentation cropping step uses a higher portion of the initial image than in CIFAR-10: we crop images of size 224x224 from images of size 256x256.

## F  Effect of number of Monte Carlo samples on ImageNet

In the paper, we report results for training ResNet-18 on ImageNet using 128 GPUs, with 1 independent Monte-Carlo (MC) sample per process during training (`mc=128x1`), and 10 MC samples per validation image (`val_mc= 10`). We now show that increasing either of training or testing MC samples improves performance (validation accuracy and log likelihood) at the cost of increased computation time. See Figure 11.

Increasing the number of training MC samples per process reduces noise during training. This effect is observed when training on CIFAR-10, where multiple MC samples per process are required to stabilise training. On ImageNet, we have much larger minibatch size (4,096 instead of 256) and more parallel processes (128 not 8), and so training with 1 MC sample per process is still stable. However, as shown in Figure 11, increasing the number of training MC samples per process to from 1 to 2 speeds up convergence per epoch, and reaches a better converged solution. The time per epoch (and hence total runtime) also increases by approximately a factor of 1.5. Increasing the number of train MC samples per process to 3 does not increase final test performance significantly.

Increasing the number of testing MC samples from 10 to 100 (on the same trained model) also results in better generalisation: the train accuracy and log likelihood are unchanged, but the validation accuracy and log likelihood increase. However, as we run an entire validation on each epoch, increasing validation MC samples also increases run-time.

These results show that, if more compute is available to the user, they can improve VOGN's performance by improving the MC approximation at either (or both) train-time or test-time (up to a limit).

## G  MC-dropout's sensitivity to dropout rate

We show MC-dropout's sensitivity to dropout rate, $p$, in this Appendix. We tune MC-dropout as best as we can, finding that $p = 0.1$ works best for all architectures trained on CIFAR-10 (see Figure 12 for the dropout rate's sensitivity on LeNet-5 as an example). On ResNet-18 trained on ImageNet, we find that MC-dropout is extremely sensitive to dropout rate, with even $p = 0.1$ performing badly. We therefore use $p = 0.05$ for MC-dropout experiments on ImageNet. This high sensitivity to dropout rate is an issue with MC-dropout as a method.

Figure 12: Effect of changing the dropout rate in MC-dropout, training LeNet-5 on CIFAR-10. When $p = 0.01$, the train-test gap on accuracy and log likelihood is very high (10.3% and 0.34 respectively). When $p = 0.1$, gaps are 1.4% and 0.04 respectively. When $p = 0.2$, the gaps are -7.71% and -0.02 respectively. We therefore choose $p = 0.1$ as it has high accuracy and log likelihood, and small train-test gap.

Figure 13: Effect of changing the dropout rate in MC-dropout, training Resnet-18 on ImageNet. We use $p = 0.05$ for our results.

## H Uncertainty metrics

We use several approaches to compare uncertainty estimates obtained by each optimiser. We follow the same methodology for all optimisers: first, tune hyperparameters to obtain good accuracy on the validation set. Then, test on uncertainty metrics. For multi-class classification problems, all of these are based on the predictive probabilities. For non-Bayesian approaches, we compute the probabilities for a validation input $\mathbf{x}_i$ as $\hat{p}_{ik} := p(y_i = k|\mathbf{x}_i, \mathbf{w}_*)$, where $\mathbf{w}_*$ is the weight vector of the DNN whose uncertainty we are estimating. For Bayesian methods, we can compute the predictive probabilities for each validation example $\mathbf{x}_i$ as follows:

$$\hat{p}_{ik} := \int p(y_i = k|\mathbf{x}_i, \mathbf{w})p(\mathbf{w}|\mathcal{D})d\mathbf{w} \approx \int p(y_i = k|\mathbf{x}_i, \mathbf{w})q(\mathbf{w})d\mathbf{w} \approx \frac{1}{C}\sum_{c=1}^{C} p(y_i = k|\mathbf{x}_i, \mathbf{w}^{(c)}),$$

where $\mathbf{w}^{(c)} \sim q(\mathbf{w})$ are samples from the Gaussian approximation returned by a variational method. We use 10 MC samples at validation-time for VOGN and MC-dropout (the effect of changing number of validation MC samples is shown in Appendix F). This increases the computational cost during testing for these methods when compared to Adam or SGD.

Using the estimates $\hat{p}_{ik}$, we use three methods to compare uncertainties: validation log loss, AUROC and calibration curves. We also compare uncertainty performance by looking at model outputs when exposed to out-of-distribution data.

**Validation log likelihood.** Log likelihood (or log loss) is a common uncertainty metric. We consider a validation set of $N_{Va}$ examples. For an input $\mathbf{x}_i$, denote the true label by $\mathbf{y}_i$, a 1-of-$K$ encoded vector with 1 at the true label and 0 elsewhere. Denote the full vector of all validation outputs by $\mathbf{y}$. Similarly, denote the vector of all probabilities $\hat{p}_{ik}$ by $\mathbf{p}$, where $k \in \{1, ..., K\}$. The validation log likelihood is defined as $\ell(\mathbf{y}, \hat{\mathbf{p}}) := \frac{1}{N_{Va}} \sum_{i=1}^{N_{Va}} \sum_{k=1}^{K} y_{ik} \log \hat{p}_{ik}$.

Tables 1 and 3 show final validation (negative) log likelihood. VOGN performs very well on this metric (aside from on CIFAR-10/AlexNet, with or without DA, where MC-dropout performs the best). All final validation log likelihoods are very similar, with VOGN usually performing similarly to the other best-performing optimisers (usually MC-dropout).

**Area Under ROC curves (AUROC).** We consider Receiver Operating Characteristic (ROC) curves for our multi-way classification tasks. A potential way that we may care about uncertainty measurements would be to discard uncertain examples by thresholding each validation input's predicted class' softmax output, marking them as too ambiguous to belong to a class. We can then consider the remaining validation inputs to either be correctly or incorrectly classified, and calculate the True Positive Rate (TPR) and False Positive Rate (FPR) accordingly. The ROC curve is summarised by its Area Under Curve (AUROC), reported in Table 1. This metric is useful to compare uncertainty performance in conjunction with the other metrics we use. The AUROC results are very similar between optimisers, particularly on ImageNet, although MC-dropout performs marginally better than the others, including VOGN. On all but one CIFAR-10 experiment (AlexNet, without DA), VOGN performs the best, or tied best. Adam performs the worst, but is surprisingly good in CIFAR-10/ResNet-18.

**Calibration Curves.** Calibration curves [7] test how well-calibrated a model is by plotting true accuracy as a function of the model's predicted accuracy $\hat{p}_{ik}$ (we only consider the predicted class' $\hat{p}_{ik}$). Perfectly calibrated models would follow the $y = x$ diagonal line on a calibration curve. We approximate this curve by binning the model's predictions into $M = 20$ bins, as is often done. We show calibration curves in Figures 1 and 14. We can also consider the **Expected Calibration Error (ECE)** metric [40, 15], reported in Table 1. ECE calculates the expected error between the true accuracy and the model's predicted accuracy, averaged over all validation examples, again approximated by using $M$ bins. Across all datasets and architectures, with the exception of LeNet-5 (which we have argued causes underfitting), VOGN usually has better calibration curves and better ECE than competing optimisers. Adam is consistently over-confident, with the calibration curve below the diagonal. Conversely, MC-dropout is usually under-confident, with too much noise, as mentioned earlier. The exception to this is on ImageNet, where MC-dropout performs well: we excessively tuned the MC-dropout rate to achieve this (see Appendix G).

# I  Out-of-distribution experimental setup and additional results

We use experiments from the out-of-distribution tests literature [16, 31, 8, 32], comparing VOGN to Adam and MC-dropout. Using trained architectures (LeNet-5, AlexNet and ResNet-18) on CIFAR-10, we test on SVHN, LSUN (crop) and LSUN (re-size) as out-of-distribution datasets, with the in-distribution data given by the validation set of CIFAR-10 (10,000 images). The entire training set of SVHN (73,257 examples, 10 classes) [42] is used. The test set of LSUN (Large-scale Scene UNderstanding dataset [55], 10,000 images from 10 different scenes) is randomly cropped to obtain LSUN (crop), and is down-sampled to obtain LSUN (re-size). These out-of-distribution datasets have no similar classes to CIFAR-10.

Similar to the literature [16, 30], we use 3 metrics to test performance on out-of-distribution data. Firstly, we plot histograms of predictive entropy for the in-distribution and out-of-distribution datasets, seen in Figure 5, 15, 16 and 17. Predictive entropy is given by $\sum_{k=1}^{K} -\hat{p}_{ik} \log \hat{p}_{ik}$. Ideally, on out-of-distribution data, a model would have high predictive entropy, indicating it is unsure of which class the input image belongs to. In contrast, for in-distribution data, good models should have many examples with low entropy, as they should be confident of many input examples' (correct) class. We also compare AUROC and FPR at 95% TPR, also reported in the figures. By thresholding the most

likely class' softmax output, we assign high uncertainty images to belong to an unknown class. This allows us to calculate the FPR and TPR, allowing the ROC curve to be plotted, and the AUROC to be calculated.

We show results on AlexNet in Figure 15 and 16 (trained on CIFAR-10 with DA and without DA respectively) and on LeNet-5 in Figure 17. Results on ResNet-18 is in Figure 5. These results are discussed in Section 4.2.

## J    Author contributions statement

List of Authors: Kazuki Osawa, Siddharth Swaroop, Anirudh Jain, Runa Eschenhagen, Richard E. Turner, Rio Yokota, Mohammad Emtiyaz Khan.

M.E.K., A.J., and R.E. conceived the original idea. This was also discussed with R.Y. and K.O. and then with S.S. and R.T. Eventually, all authors discussed and agreed with the main focus and ideas of this paper.

The first proof-of-concept was done by A.J. using LeNet-5 on CIFAR-10. This was then extended by K.O. who wrote the main PyTorch implementation, including the distributed version. R.E. fixed multiple issues in the implementation, and also pointed out an important issue regarding data augmentation. S.S., A.J., K.O., and R.E. together fixed this issue. K.O. conducted most of the large experiments (shown in Fig. 1 and 4). The results shown in Fig. 3a was done by both K.O. and A.J. The BBB implementation was written by S.S.

The experiments in Section 4.2 were performed by A.J. and S.S. The main ideas behind the experiments were conceived by S.S., A.J., and M.E.K. with many helpful suggestions from R.T. R.E. performed the permuted MNIST experiment using VOGN for the continual-learning experiments, and S.S. obtained the baseline results for the same.

The main text of the paper was written by M.E.K. and S.S. The section on experiments was first written by S.S. and subsequently improved by A.J., K.O., and M.E.K. R.T. helped edit the manuscript. R.E. also helped in writing parts of the paper.

M.E.K. led the project with a significant help from S.S.. Computing resources and access to the HPCI systems were provided by R.Y.

## K    Changes in the camera-ready version compared to the submitted version

- We added an additional experiment on a continual learning task to show the effectiveness of VOGN (Figure 3b).
- In our experiments, we were using a damping factor $\gamma$. This was unfortunately missed in the submitted version, and we have now added it in Section 3.
- We modified the notation for Noisy K-FAC algorithm at Appendix A.
- We updated the description of our implementation of the Gauss-Newton approximation at Appendix B. Previous description had some missing parts and was a bit unclear.
- We added a description on a new method OGN which we were using to tune hyperparameters of VOGN. We have added its results in Table 1 and Table 3. The method details are in Appendix C.
- We added a description on how to tune VOGN to get good performance.
- We listed all training curves (epoch/time vs accuracy), including K-FAC, Noisy K-FAC, and OGN, along with the corresponding calibration curves in Figure 7.

(a) LeNet-5 on CIFAR-10 (no DA)

(b) AlexNet on CIFAR-10 (no DA)

(c) AlexNet on CIFAR-10

(d) ResNet-18 on CIFAR-10

(e) ResNet-18 on ImageNet

Figure 7: All results in Table 1

Figure 8: Effect of prior variance on VOGN training ResNet-18 on ImageNet.

Figure 9: Effect of prior variance on Noisy K-FAC training ResNet-18 on ImageNet.

Figure 10: Effect of changing the dataset size reweighting factor $\rho$ and prior variance on VOGN training ResNet-18 on ImageNet.

Figure 11: Effect of number of training and testing Monte Carlo samples on validation accuracy and log loss for VOGN on ResNet-18 on ImageNet.

Figure 14: Calibration curves comparing VOGN, Adam and MC-dropout for final trained models trained on CIFAR-10. VOGN is extremely well-calibrated compared to the other two optimisers (except for LeNet-5, where all optimisers peform well). The calibration curve for ResNet-18 trained on ImageNet is in Figure 1.

Figure 15: Histograms of predictive entropy for out-of-distribution tests for AlexNet trained on CIFAR-10 with data augmentation. Going from left to right, the inputs are: the in-distribution dataset (CIFAR-10), followed by out-of-distribution data: SVHN, LSUN (crop), LSUN (resize). Also shown are the AUROC metric (higher is better) and FPR at 95% TPR metric (lower is better), averaged over 3 runs. The standard deviations are very small and so not reported here.

Figure 16: Histograms of predictive entropy for out-of-distribution tests for AlexNet trained on CIFAR-10 without data augmentation. Going from left to right, the inputs are: the in-distribution dataset (CIFAR-10), followed by out-of-distribution data: SVHN, LSUN (crop), LSUN (resize). Also shown are the AUROC metric (higher is better) and FPR at 95% TPR metric (lower is better), averaged over 3 runs. The standard deviations are very small and so not reported here.

Figure 17: Histograms of predictive entropy for out-of-distribution tests for LeNet-5 trained on CIFAR-10 without data augmentation. Going from left to right, the inputs are: the in-distribution dataset (CIFAR-10), followed by out-of-distribution data: SVHN, LSUN (crop), LSUN (resize). Also shown are the AUROC metric (higher is better) and FPR at 95% TPR metric (lower is better), averaged over 3 runs. The standard deviations are very small and so not reported here.

## Footnotes

[5]`https://pytorch.org/docs/stable/nn.functional.html#torch.nn.functional.unfold`

[6]`https://pytorch.org/tutorials/beginner/former_torchies/nnft_tutorial.html#forward-and-backward-function-hooks`

[7]This gradient approximation used here is referred to as the *zeroth-order delta approximation* where $\mathbb{E}_q[\hat{\mathbf{g}}(\mathbf{w})] \approx \hat{\mathbf{g}}(\boldsymbol{\mu})$ (see Appendix A.6 in Khan [21] for details).