[Reviews · NeurIPS 2019]

Reviewer 1



This paper proposes a deep learning training strategy using natural gradient variational inference, and claims that this will preserve the benefits of bayesian principles. However, experimental results by the proposed method are not very impressive compared with other baselines despite the more complicated training process. In addition, I think it would be better if the author can discuss more about how easy the proposed method can be generalized to different deep models.

Reviewer 2



Originality: Rather low The main technical novelty lies in applying tricks from the deep learning literature to VOGN. The experiments are fairly standard. Quality: High That being said, the experiments seem to be carefully executed, described in detail and the overall method is technically sound. While not overly ambitious in terms of technical novelty, I think this is a well-executed piece of work. Clarity: High The paper is well-written and easy to follow. Significance: Mixed I find that the paper does itself a bit of a disservice by putting so much focus on technicalities. I believe this in an attempt to appeal to readers with an interest in deep learning rather than Bayesian inference, however I don't find the empirical part of the paper to make a particularly strong case for using Bayesian methods in deep learning. My main takeaway from the experiments would be that "being Bayesian" does not matter too much on a large dataset like Imagenet (or even CIFAR-10) and the small calibration improvements as in Figure 1 are probably not worth the extra headache. If the authors indeed wish to make a case for Bayesian deep learning to a larger audience, I think that the paper would be much stronger if it had some online learning or fine-tuning experiments when using the approximate posterior as a prior on a much smaller dataset, where ignoring parameter uncertainty would most likely lead to dramatically worse performance. The numbers in Table 1 are too close/inconsistent to be really convincing in an empirical paper and for the out-of-distribution uncertainty as in Figure 5 it is unclear if it is a good metric since we don't know the uncertainty of the true posterior. Alternatively, this could also be a much more relevant contribution to the Bayesian deep learning subfield if the paper made an attempt to gain insight into why VOGN works better than e.g. BBB. The paragraph in lines 91 to 97 does not make much sense to me, since (unless I misunderstood something) both methods optimize the same objective with an approximate posterior from the same parametric family - the difference is that VOGN is a natural gradient method. So the failure of BBB can't be attributed to the ELBO if VOGN works. But if the argument is that natural gradient is necessary, I find it surprising that Nosiy KFAC is apparently difficult to tune. Digging a bit deeper here would probably lead to interesting insights.

Reviewer 3



This paper proposes a perspective on training Bayesian Neural Networks (BNNs) that motivates how to best incorporate different tricks (such as batch normalization and momentum) into BNN training. The resulting algorithm scales to large inference problems like fitting a BNN to ImageNet and achieves well calibrated predictions. Starting point is an existing approach (VOGN) for fitting BNNs with natural gradients. The authors observe that the update equations of VOGN are similar to the update equations of popular SGD methods with adaptive learning rates. From this perspective, they can derive by analogy how to best incorporate different tricks for practical deep learning (batch normalization, data augmentation, distributed training). The extensive experimental study supports the claims of the authors. Topic-wise, this work is a good fit to the Neurips community. There seem to be no 'new ideas' in this paper (VOGN comes from ref [22] and batch normalization, data augmentation, etc. come from the deep learning literature), so I would rate it lower on originality. Yet, I find it an important contribution to bridging the gap between Bayesian neural networks and practical deep learning. The ideas and how they are connected are described clearly. This work is an interesting step into the direction of finding the right trade-off between computational efficiency and well calibrated predictions in Bayesian deep learning.

[Author Response · NeurIPS 2019]

We would like to thank the reviewers for their feedback and time.

**Significance of the work.** R2 and R3 are concerned about the significance of the work. We respectfully disagree with
the opinion that the significance is low/mixed. The Bayesian deep learning (BDL) community, which is our target
audience, struggles to obtain good performance with VI methods on large problems such as ImageNet (see [3] as an
example). This paper is the first to close this gap (as R3 and R5 both mention). Our codebase will enable the community
to easily apply VI on large problems, which is a significant contribution. There is a misunderstanding among reviewers
regarding our target audience, and we will modify the introduction to make sure that this is fixed.

**Significance of experimental results.** R2 and R3 find the experiments not to be convincing and state that our methods
do not beat the baselines. We emphasise that this paper is not about beating the state-of-the-art, rather it is about
showing that a principled approach works well. The BDL community currently largely relies on MC-dropout, and our
goal is to show them that VI can achieve similar or better performance.

Our results achieve this objective. Results in Table 1 show that our method performs comparably to SGD, Adam, and
MC-dropout. Calibration curves (in Fig. 1) and OOD tests (in Fig. 5) show that uncertainty performance is better than
MC-dropout and Adam. It is true that there is no clear winner in Table 1, which is perhaps the reason behind reviewers'
concerns. But VOGN does provide a marginally better performance, e.g., on CIFAR-10, on uncertainty metrics, VOGN
is consistently either best or tied best (8 out of 12 numbers) or else second best, while both Adam and MC-dropout vary
wildly. We will modify the text to make these points clear.

**Additional experiments.** R2 and R3 ask for more experiments to
show the benefits of Bayesian principles. We will add two experi-
ments in the paper (or in Appendices, depending on space constraints):
(i) a continual learning experiment, (ii) the Diabetic Retinopathy Di-
agnosis benchmark for Bayesian models [5] (this benchmark has
only recently been released). For (i), we compare with EWC [4] and
Variational Continual Learning (VCL) [1] on 10 tasks of permuted
MNIST, using the same architecture and setup as in the VCL paper.
Part of the results are shown in Fig. 1 where VOGN achieves $94\pm1\%$

Figure 1: VOGN clearly outperforms EWC (curve from [2]) on 10 tasks of permuted MNIST.

(20 runs), which is better than EWC and SI [1], and marginally better
than VCL's performance of $93\pm1\%$ [2]. An advantage of VOGN over VCL is that it is much faster to converge.

**R2: "discuss more about how easy the proposed method can be generalized to different deep models."** VOGN
is a plug-and-play optimiser in PyTorch that is very easy to use (see lines 46-58 in `utils.py`, Supplementary material).
Applying VOGN, instead of Adam, requires just 2-3 lines of code change.

**R3: "for the out-of-distribution uncertainty ... we don't know the uncertainty of the true posterior."** The true
posterior is never available in such complex settings, and many other papers have focused on metrics for out-of-
distribution uncertainty (see the references in lines 257-258 in the paper).

**R3: "... an attempt to gain insight into why VOGN works better than e.g. BBB."** Thank you for raising this point.
As noted by R5, the main reason why VOGN works is the similarity of its updates to Adam, which makes it easier to
apply the performance-improvement techniques used in deep-learning. We will add more discussion in the paper so
that this point is clearly communicated. We do find that BBB is accurate whenever we can get it to work, but then it is
extremely slow to converge; VOGN on the other hand is much quicker (see results on CIFAR-10/LeNet-5). Applying
similar techniques for BBB does not work since the updates are very different from Adam (this is what we are saying in
lines 91-97: we will modify the text to improve clarity). We have tried many tricks on BBB, including using the local
reparameterisation trick and suitably initialising means and variances (as recommended by [2]). We did not specifically
try the trick you mentioned.

**R3: "I find it surprising that Noisy KFAC is apparently difficult to tune."** What we meant is that Noisy K-FAC is
much slower than VOGN, which makes it difficult to find good hyperparameter settings.

**R5: "how your approach compares to stochastic gradient langevin dynamics."** In VOGN, weights of the neural
network are perturbed, while in preconditioned SGLD, gradients are perturbed. If it helps, we can add a simulation
comparing the two (although this type of comparison is done previously in [6]).

[1] C.V. Nguyen et al. Variational continual learning. ICML, 2018.
[2] S. Swaroop et al. Improving and understanding variational continual learning. arXiv:1905.02099, 2019.
[3] Y. Ovadia et al. Can You Trust Your Model's Uncertainty? Evaluating Predictive Uncertainty Under Dataset Shift.
arXiv:1906.02530, 2019.
[4] J. Kirkpatrick et al. Overcoming catastrophic forgetting in neural networks. PNAS, 2017.
[5] A. Filos et al. Benchmarking Bayesian Deep Learning with Diabetic Retinopathy Diagnosis. 2019.
[6] Z. Nado et al., Stochastic gradient langevin dynamics that exploit neural network structure. ICLR, 2019.


[Meta-Review · NeurIPS 2019]

The paper demonstrates that the Variational Online Gauss-Newton (VOGN) method of Khan et al. (2018) can be successfully scaled to deep learning architectures. The authors demonstrated the scalability of Bayesian methods to large scale data such as ImageNet. Extensive experiments on large scale data and models are provided. The main result is an adoption of an existing model (VOGN) to make it practical for deep learning.